# Digital Twins Solve the Mystery of Raman Spectra of Parental and Reduced Graphene Oxides

**DOI:** 10.3390/nano12234209

**Published:** 2022-11-26

**Authors:** Elena F. Sheka

**Affiliations:** Institute of Physical Researches and Technology, Peoples’ Friendship University of Russia (RUDN University), 117198 Moscow, Russia; sheka@icp.ac.ru

**Keywords:** digital twins concept, virtual vibrational spectrometry, Hartree-Fock approximation, IR and Raman spectra, reduced graphene oxide, graphene oxide, *sp*^2^-to-*sp*^3^ bond transformation

## Abstract

Digital Twins concept presents a new trend in virtual material science, common to all computational techniques. Digital twins, virtual devices and intellectual products, presenting the main constituents of the concept, are considered in detail on the example of a complex problem, which concerns an amazing identity of the D-G-doublet Raman spectra of parental and reduced graphene oxides. Digital twins, presenting different aspects of the GO and rGO structure and properties, were virtually synthesized using a spin-density algorithm emerging from the Hartree-Fock approximation. Virtual device presents AM1 version of the semi-empirical unrestricted HF approximation. The equilibrium structure of the twins as well as virtual one-phonon harmonic spectra of IR absorption and Raman scattering constitute a set of intellectual products. It was established that in both cases the D-G doublets owe their origin to the *sp*^3^ and *sp*^2^ C-C stretchings, respectively. This outwardly similar community reveals different grounds. Thus, multilayer packing of individual rGO molecules in stacks provides the existence of the *sp*^3^ D band in addition to *sp*^2^ G one. The latter is related to stretchings of the main pool of *sp*^2^ C-C bonds, while the *sp*^3^ constituent presents out-of-plane stretchings of dynamically stimulated interlayer bonds. In the GO case, the *sp*^3^ D component, corresponding to stretchings of the main pool of *sp*^3^ C-C bonds, is accompanied by an *sp*^2^ G component, which is related to stretchings of the remaining *sp*^2^ C-C bonds provided with the spin-influenced prohibition of the 100% oxidative reaction in graphene domain basal plane.

## 1. Introduction

The essence of the DTs concept concerns the trinity of physical objects, virtual/digital objects, and the connection between them (see [1,2] and references therein). The latter is provided by the data that flows from the physical object to the digital/virtual one and information that is available from the digital/virtual object to the physical environment. The modelling aspect of this concept, which is new for large massive fields of human activity, has been widely exploited in academic research since the first computer became available. Known as simulations or modelling, the concept implementation has provided a drastic development in academic studies, related to natural science, in particular. Today it is impossible to imagine modern physics, chemistry, and material science (as well as they are the same but with prefixes bio- and geo-) without modelling. A huge leap in the development of computational programs and computing tools that have taken place over the past half-century has led to the fact that many previously predominantly empirical sciences have become virtual-empirical, while some of them such as graphenics occur predominantly virtual [3]. Despite such rapid development, the relationship between the real object and its model, established as a subordinate to the model, has not changed until recently. As it turned out, this circumstance significantly limits the further development of science in the case of its predominantly virtual nature. Here, the DTs concept in the above wording enters the scene. The difference between the concept and previous modelling is in a different sense, which is embedded in the understanding of the connection between physical and digital objects. Applying to a language grammar, it can be represented as a difference between complex sentence (principal-subordinate) in the case of modelling and compound one (equal-equal) related to DTs. In every language, substituting one sentence for another changes the meaning of the spoken speech. The same is true in the case of science [2].

First used recently [4], the contradistinction of compound DT concept to complex molecular modelling has revealed a high efficiency of the former to solve intricate complicated problems. It was natural to apply to it looking for reasons for the identity of Raman spectra of rGO and GO and finding a way to discover this mystery. We look at the DT concept following Scheme (1), which connects the three constituents of the approach:(1)Digital twins→ Virtual device→IT product

Here, DTs are molecular models under study, a virtual device is a carrier of selected software, IT product covers a large set of computational results related to the DTs under different actions in the light of the soft explored. The quality of the product highly depends on how broadly and deeply the designed DTs cover all the knowledge concerning the object under consideration and how adequate is the virtual device to the peculiarities of the object under study. The first requirement can be evidently met by a large set of the relevant models, in the current case up to a few hundred, each consisting of *N* atoms (up to hundreds). As for the virtual device, it should not contradict the object’s nature and perform quantum-chemical calculations providing the establishing of the equilibrium structure of the designed DTs and obtaining their spectra of IR absorption and Raman scattering related to 3*N* vibrational modes. Only semi-empirical quantum programs based on either Hartree-Fock (HF) approximations [5] or the theory of density functional (DFT) as well as classical molecular dynamics can cope with such a volume of cumbersome quantum-chemical calculations. As for quantum approaches, the radical nature of most rGO and/or GO DTs, which turns them into open-shell electronic systems, forced to abandon DFT-based programs and to pay attention to programs based on the unrestricted Hartree–Fock approximation. In the performed studies [4], the virtual device presents software CLUSTER-Z1 that realizes both restricted (RHF) and unrestricted (UHF) versions of the semi-empirical AM1 HF approximation described elsewhere in details [6] as well as provides one-phonon harmonic spectra calculations [7].

## 2. The Problem of Identity of Raman Spectra of Parental and Reduced Graphene Oxides

Currently, graphene oxide (GO) and reduced graphene oxide (rGO) top the list of high-tech modern graphene material science, which is explained by the relatively easy manufacturing, which tightly couples these materials, and their moderate cost. The list of publications is practically countless as can be seen from the recent reviews (see [8,9,10,11,12,13,14,15,16], but a few). Belonging to an extensive family of solid carbons, GO and rGO live their lives quite separately. As occurred, GO does not exist in nature and is a synthetic product—graphite oxide known since 1859 [17]. In contrast, rGO exists in nature for many millions of years in deposits of diverse *sp*^2^ amorphous carbons, including various coals, shungite carbons, anthraxolites, and accompanying carbon sheaths-shells of many other minerals [18]. However, the attribution of this richness to that of rGO has been realized only recently, when it was found that all the *sp*^2^ amorphous carbons listed above are multilevel structures based on basic structural units (BSUs), which are graphene domains surrounded with necklaces of heteroatoms. This presentation of necklaced graphene molecules suits exactly the manufactured rGO as well, solids of which are just such a multilevel amorphous structure. Appealing to synthetic manufacturing [19,20,21,22,23,24,25,26], modern graphenics classifies GO and rGO as an almost inseparable pair.

One of the motivations inclining to the acceptance of this internal kinship inseparability is the hitherto unsolved riddle of the amazing identity of the Raman spectra of GO and rGO. If we take into account that the carbon carcass of these two covalent solids has a completely different structure, which is a net of condensed non-planar ‘puckered’ cyclohexanoid units of GO in contrast to flat benzenoid ones of rGOs, then the observed identity becomes a unique spectral phenomenon, never observed before for covalent molecules and solids. At the same time, the high demand for both materials in modern high-tech production forces us to unravel this mystery, which gives rise to great uncertainty in understanding the internal structure of both materials, which evidently prevents their optimal use. This paper is aimed at solving the task of attempting to answer the main question—why Raman spectra of the two carbon oxides are so similar. It is precisely the lack of understanding of this exceptional oddity that forces material scientists to operate with a complex “identity certificate” of these materials, figuratively presented in Figure 1 and based on a comparison of the properties of GO and rGO. As clearly seen, the structure and chemical composition of GO and rGO are completely different and the only characteristic, which is the same for both substances, is their Raman spectra.

Raman spectra are widely used for structural and chemical analysis of substances, particularly covalent ones, precisely because of their dependence on both characteristics. Naturally, the observed exotic identity could not fail to attract the attention of researchers. However, until recently this unique circumstance has not received a convincing explanation [27,28,29,30]. Actually, numerous Raman spectra of various graphene-based materials convincingly testify the presence of a characteristic D-G-band-doublet signature (see review [31] and references therein), which together with independent evidence of planar graphene domains in the ground of the relevant carbon carcass allowed making undeniable verdict on the graphene nature of the bodies attributed as rGOs. Simultaneously, numerous efforts have been made to preserve the identification ability of this spectral mark, which led to giving an exclusive role to two parameters of the spectra, namely, to the ratio of the intensity of the D and G bands, IDIG, and the corresponding bands half-widths, Δω, to characterize the size and defect structure of the relevant graphene domains. This relationship was established theoretically for graphene crystal [32,33] and then transferred to nanoscale rGOs of amorphous substances [34,35]. However, as was shown lately [31], such a transfer turned out to be incorrect, which, nevertheless, has not stopped the efforts of the “theoretical description of the defectiveness” of the studied rGOs until now and has been accepted for GO by default.

The next reason for keeping the identity of the Raman spectra of GO and rGO uncovered is closely related to the first launch of the mandatory proof of the “graphene-domain origin” of the Raman spectrum of GO (see review [28] and references therein). This formulation of the problem was stimulated by the presence of ‘zones of the original graphite that did not undergo oxidation’ among the solid GO massive observed experimentally [36,37]. Thus, a simplified idea appeared about weak interaction of graphene domains in GO with groups of atoms containing oxygen [38], which drastically contradicts the grounds of organic chemistry. In turn, the remained problem is the cause of great confusion in the designation of the material used in a large number of articles devoted to ‘graphene’ application in the field of material science associated with various fields of chemical and biochemical technologies, as is, say, the case of Covid vaccines [39].

## 3. Spin-Density Algorithm of the Digital Twins Design

Both rGO and GO are polyderivatives of bare graphene domains, but of a different class. rGOs are related to derivatives resulted from the oxidation reactions that occurred on the domain edge atoms only and present a large family of necklaced graphene molecules with *sp*^2^ configured basal plane atoms. GOs are products of the derivatization that includes not only edge carbon atoms of the domain, but basal-plane ones as well. Once fully derivatized, both edge and basal-plane *sp*^2^ carbon atoms of the pristine domains become *sp*^3^ hybridized. Basal-plane oxidation results in a severe transformation of the previously flat benzenoid structure thus transferring the latter to a puckered cyclohexanoid one.

From the vibrational dynamic viewpoint, benzenoid and cyclohexanoid vibrational signatures are absolutely different [40,41]. Additionally, the cyclohexanoid structures are subjected to a large conformity, which greatly complicates both the prediction of the post-derivatization structure and the vibrational analysis of the latter. One of the ways to clarify what is going on with the pristine domain under edge-and-basal derivatization is to trace subsequent steps of the reaction one by one. Computationally, it can be carried out by applying the spin molecular theory of *sp*^2^ nanocarbons [42], perfectly demonstrating its efficacy by the virtual design of the graphene derivatives such as hydrides and oxides as well as polyderivatives of fullerene C_60_ under stepwise fluorination, hydrogenation, and so forth. The virtual synthesis was governed by a spin-density algorithm, in frame of which, the value of the atomic chemical susceptibility (ACS) is a quantitative indicator of the atom chemical activity allowing to match target for any next step of the considered reaction. The algorithm is based on the ACS strong dependence on the length of the relevant covalent C-C bond. Concerning *sp*^2^C-C ones, ACS is nil until the bond length reaches the critical value $R_{crit}$ = 1.395 Å. Usually, both empirical and calculated bond lengths of graphene domains fill the interval of 1.322–1.462 Å. The relative number of bonds, whose length exceeds $R_{crit}$, constitutes ~60% of the bond pool. The resulting radicalization leads to a considerable amount of the effectively unpaired electrons, *N_D_*. From the chemical viewpoint, the *N_D_* value describes the domain (molecule) chemical susceptibility in total, while *N_DA_* presents ACS related to the domain’s atom A. These two quantities are the main parameters of the spin-density algorithm, which governs the oxidation of graphene domains discussed in the current study.

Figure 2b exhibits a typical ACS image map presented by the $N_{DA}$ distribution over 66 atoms of the bare domain (5,5) NGr, *N_D_* of which constitutes 31 e. The molecule edges are not terminated, and the ACS map has a characteristic view with a distinct framing by edge atoms which looks like a typical ‘dangling bonds’ icon. The absolute $N_{DA}$ values of the domain, shown by light gray plotting in Figure 2e, clearly exhibit 22 edge atoms involving 2 × 5 *zg* and 2 × 6 *ach* ones with the highest $N_{DA}$ thus marking the perimeter as the most active chemical space of the molecule. The first step of any derivatization of the domain occurs on atom 14 (see star-marked light gray plotting in Figure 2e). The next step of the reaction involves the atom from the edge set as well, and this is continuing until either all the edge atoms are saturated or some of the basal ones come into play. In the case of the domain hydrogenation, the first 44 steps are accompanied by the high-rank $N_{DA}$ list where edge atoms take the first place once terminated by hydrogen pairs [43].

Thus, obtained hydrogen-necklaced graphene molecule C_66_H_44_ is shown in Figure 2c alongside the corresponding ACS image map in Figure 2d that reveals the transformation of brightly shining edge atoms in Figure 2b into dark spots. Consequently, the chemical activity is shifted to the neighboring basal atoms and retains higher near *zg* edges. The dotted curve in Figure 2e exhibits free valence distribution over the molecule atoms in the basal plane (see details in [42]). The hydrogenation of atoms in the basal plane of the necklaced (5,5) NGr starts on atom 13 (see star-marked black plotting in Figure 2e).

## 4. Digital Twins of the First Approach

### 4.1. Virtual Design

Three GOs, thus synthesized by our team [43] early, form the basis of the DTs of the first approach. Their equilibrium structures are shown in Figure 3. The basic domain (5,5) NGr is of 1.2 × 1.1 nm^2^ in size and suits well not only empirical necklaced-graphene compositions of rGO related to *sp*^2^ amorphous carbons [31], but to *sp*^3^ structural units of GOs of ~2 nm^2^ in size [44] as well. Nothing to say that the identity of the graphene domain in the ground of the DTs, related to both GO and rGO, is an exceptional favoring bonus for the current study.

Pronounced conformation of cyclohexanoid compositions causes the dependence of the formed structure on such external factors as the fixation of the graphene domain edges, the accessibility of the domain basal plane to heteroatoms participating in the reaction from one or both sides (*up* and *down* formats). In what follows, particular marking subscriptions *fx*, *fr*, *1*, and *2* will be used to distinguish external conditions of the DTs design. The set of GOs in Figure 3 forms a series of *fr2*, *fr1*, and *fx2* DTs indicating that the graphene domain edges of GO1 and GO2 were free-standing while those of GO3 were fixed. Simultaneously, the basal plane in the case of GO1 and GO3 was accessed *up* and *down*, while in the GO2 case the accession was one-side only. Moreover, the virtual synthesis was complicated with the difference in oxygen-containing units involved. GO1 and GO2 are the results of oxidation in the concurrent presence of atomic oxygens, hydroxyls, and carboxyls, while GO3 presents a product, virtually synthesized in a flow of hydroxyls only.

The design of GO1 and GO2 started at the domain edge atoms, which was accompanied by a successive consideration of the addition of single oxygen atoms, hydroxyls, and carboxyls at each step. After evaluating the binding energy (BE) of each attachment separately, the choice of the obtained derivatives was made in favor of the configuration with the highest BE. This process continued 22 steps, because of which the necklaced graphene molecule C_66_O_22_ was formed, and the oxidation process moved to the basal plane. Each step of this process was considered as a choice between the involvement of an *sp*^2^C-C bond in the formation of either a C_2_O epoxy group or opening the bond with the addition of two hydroxyl groups, thus forming C_2_(OH)_2_ composition. In addition, the landing of heteroatoms *up* or *down* in the case of GO1 and only *up* in the case of GO2 was controlled. The oxidation process was stopped at the 17th step concerning basal-plane atoms due to zeroing the corresponding $N_{D}$ and $N_{DA}\;$values. In both cases, the basal plane, consisting of 44 carbon atoms, is covered with 14 C_2_O epoxy and 2 C_2_(OH)_2_ groups. The virtual synthesis of GO3 was performed with the participation of hydroxyls only. At the first stage of the reaction, a C_66_(OH)_36_ necklaced graphene molecule was synthesized. After moving the reaction to the basal plane, another 38 OH groups were added *up* and *down* until both $N_{D}$ and $N_{DA}\;$were zeroing. All the above derivatization included about 400 computational jobs [43].

The virtual design showed that carboxyls are characterized by the least BE at each step of the oxidation due to which their presence in the basal plane area is unfavorable. As for the graphene domain circumference, some single attachments might be possible under particular conditions of strong perturbation of the structure. This conclusion is supported by GO experimental structural data. As shown, solid GO consists of layered stacks of a few nm in thickness and of 2 nm^2^ in lateral dimension [36,44]. The interlayer distance drastically varies because of the water contamination, caused by the solid’s high hydrophilicity, and the minimal value constitutes 0.784 to 11.xx nm [45,46]. If take into account that the thickness of a puckered cyclohexanoid is bigger than that of flat benzenoid on 0.124 nm, carbon atoms take on 0.459 nm of distance. The remaining 0.325 nm of the minimal distance allows the location of either oxygen atoms (0.304 nm [47]) of epoxy groups or bent hydroxyls between the layers. The interlayer distance is rather small as well to comfortably house carboxyls in the circumference of the rGO basic units. The groups are quite cumbersome and their comfortable disposition around the graphene domain requires a lot of space (see DT IV structure in [4]). In accordance with this, carboxyls will not be considered concerning the following DTs design.

### 4.2. Virtual Vibrational Spectra

The virtual device HF Specrodyn provides the calculation of harmonic one-phonon spectra of IR absorption and Raman scattering of preliminary structurally optimized DTs [48]. The calculations are based on the standard rigid-rotor harmonic-oscillator model in the framework of the semi-empirical Hartree–Fock quantum chemical approach. All the calculations, discussed in the current paper, were performed using either UHF or RHF versions of the code depending on the radicalization extent of the studied DTs. Throughout the paper, the virtual spectra are presented by stick-bars convoluted with Gaussian bandwidth of 10 cm^−1^. Intensities, reported in arbitrary units, are normalized per maximum values within each spectrum. Since the number of vibrational modes, composing the spectra under consideration, is large, the excessive fine structure, statistically suppressed in practice, is covered by trend lines.

Virtual vibrational spectra of the DTs, discussed above, are shown in Figure 4. As seen in the figure, both IR and Raman spectra of GO1 and GO2 are well similar and considerably differ from those ones related to GO3. Thereby, one- or two-side modes of the basal plane filling do not influence the IR spectra much, while the substitution of oxygen atoms with hydroxyls drastically changes these spectra. The subsequent description of IR virtual spectra is based on the general frequency kits (GFKs) presented in Table 1. The table includes GFKs previously adapted for *sp*^2^-configured rGOs [4], while supplemented with the data proposed for *sp*^3^ GO on the basis of experimental spectra [49,50,51,52,53]. The analysis was carried out on a virtual frequency scale unless otherwise indicated. Thus, IR spectra of GO1 and GO2 at 2100 cm^−1^ are presented with the *ν sp*^3^C=O modes, while those of *ν sp*^3^C-OH, *ν sp*^3^C-O-C, and *δ ip sp*^3^C-OH comparatively equally contribute to the region of 1200–1700 cm^−1^. In the GO3 IR spectrum, the main role is expectedly assigned to vibrational modes involving hydroxyls, which are presented with *δ op sp*^3^C-OH mode in the region of 400–1000 cm^−1^ as well as with a large set om modes, including *δ ip sp*^3^C-OH, *ν sp*^3^C-OH, *ν sp*^3^C-O-C, *ν sp*^3^C-C, and *ν sp*^2^C-C ones in the region of 1200–1800 cm^−1^. As seen from the table, in contrast to rGO, the *ν sp*^2^C-C modes pool of which is distinctly separated by frequency from other modes, thus promoting a distinct distinguishing of contributions of necklace heteroatoms and basal-plane carbon ones into IR and Raman spectra, respectively [4], the *ν sp*^3^C-C modes of GO overlap closely with others, thus excluding a similar distinguishing.

Looking at the Raman spectra of DTs GO1 and GO2 in Figure 4b, one sees a triplet of intense bands at ~1760 cm^−1^ (I), ~1900 cm^−1^(II), and ~2100 cm^−1^ (III). According to the GFKs listed in Table 1, the bands should be attributed to the *ν sp*^3^C-C, *ν sp*^2^C-C, and *ν sp*^3^C=O modes, respectively. Only band II of the triplet is observed in the spectrum of GO3. The *ν sp*^3^C=O origin of band III makes it possible to estimate the unavoidable shift of virtual frequencies in this region with respect to the experimental ones [55]. According to the data listed in Table 1, the blue shift is 200–300 cm^−1^. The largest value makes it possible to obtain a better agreement with the experimental data for the *ν sp*^3^C-O-H band at 3400 cm^−1^, so that we take this value as a requested one to fit virtual spectra and experimental spectra of a real sample produced by AkKo Lab [54]. As can be seen in Figure 4, shifting leads to an obvious agreement between the calculated and experimental spectra concerning both IR absorption and Raman scattering. In the latter case, the characteristic empirical doublet of D-G bands conveniently covers virtual bands I–II, making it possible to consider the latter as DTs of the D-G ones. Naturally, we are not talking about exact replicas, but about the agreement in the main elements of the structure, which allows their interpretation. Taking into account the amorphous nature of GO, one should expect the appearance of only the most significant structural elements in the experimental spectra, positioned at a considerable structureless background. Indeed, usually, the experimental spectra have just such a character [27,54]. A general consistency of experimental and virtual IR spectra in Figure 4a alongside with a large set of GFKs listed in Table 1 allows making a confident detailed analysis of the IR spectrum structure of any real GO sample.

## 5. Monochrome Digital Twins of GO of the Second Approach

Chemical content and linear dimensions of GO1 and GO2 are well consistent with experimental data, thus providing their attribution to the basic structural units of real GO samples. Virtual Raman spectra of the latter, presented in Figure 4b, confirm the validity of these models implying that particular peculiarities of vibrational spectra of GO1 and GO2 do provide the characteristic D-G doublet Raman spectrum of real GO. However, the described oxygen status of GO1 and GO2 is complicated, too ‘polychromic’, and does not allow making any conclusions about the discussed features beyond speculations. Evidently, a ‘monochromization’ of the oxide composition may be a logical step towards overcoming this polychrome difficulty.

As said earlier, DTs related to GO consist of two structural parts concerning the oxygen content. The first involves the necklace atoms, linked to the edge atoms of the parental graphene domain and responsible for the *sp*^3^ hybridization of the latter. The second part concerns configurations on the domain basal plane. Assuming that the influence of these two parts on IR and Raman spectra of GOs, as it is in the rGO case [4], is different, DTs GO4, GO5, and GO6, shown in Figure 5, were designed. Based on the graphene domain (5,5) NGr, model GO4 and GO5 differ in the necklaces (oxygen atoms providing the carbonyl one in the first case and hydrogen pairs forming the methylene necklace in the second), once covered *up* and *down* with 22 epoxy groups in both cases. Against, the necklace composition of DTs GO5 and GO6 is the same, while epoxy covering of GO5 is replaced with hydroxyl one in GO6. The virtual synthesis, based on the ACS spin-density algorithm, concerned the necklaced parts of the oxides, while *up* and *down* filling of the domain basal plane was performed by hand. In all the cases, the covering corresponds to 100% oxidation.

IR and Raman spectra of these DTs, shown in Figure 6, react with the produced monochromization in different ways. Concerning IR absorption, the replacement of the carbonyl necklace of GO4 with methylene one of GO5 and GO6 results in a drastic changing of the spectra when going from GO4 to GO5. The next transformation of GO5 into GO6 evidently conserved the methylene character of the necklace, although noticeably influenced by events occurring in the basal plane of the carbon domain. Thus, as seen in Figure 6a, IR spectra exhibit vibrations of necklacing heteroatoms mainly, influenced by the basal-plane events to some extent. In contrast, Raman spectra in Figure 6b present stretchings of the covalent C-C and C-O bonds. As seen in the figure, three types of the vibrational modes participate in the spectra formation. The first type of modes is strictly confined to the frequency region of 1000–1500 cm^−1^. We attribute them *ν sp*^3^**C**-C stretchings occurring in the environment of carbon atoms only. Actually, the modes are similar to the virtual vibrational spectrum of nanographane, based on the (5,5)NGr domain [58] (see Figure 9 below). The latter in turn is well consistent with the virtual phonon spectrum of crystalline graphane [59]. As seen in Figure 6b, this confinement really takes place in the spectra of GO4 and GO5, The next type of modes is attributed to the *ν sp*^3^C-C(O) stretchings at 1500–1800 cm^−1^ that represent *sp*^3^ C-C bonds, the environment of which involves oxygen atoms as well. A peculiar behavior of vibrational bands in organic molecules associated with carbon atoms in the vicinity of oxygens was noticed as early as 40 years ago [60,61] and since then has been constantly discussed in the literature (see [49,50] and references therein). These bands easily changed their position and intensity depending on the place of oxygen atoms in the molecule, on their number, on the coexistence of certain heteroatoms, and so forth. In the current case, the feature is well-supported by Raman spectra difference in the region between the spectra of GO4 and GO5, on the one hand, and GO6, on the other. The third type of mode concerns the *ν sp*^3^**C**=O stretchings underlying band III.

As seen in the figure, the Raman spectrum of GO4 exhibits all three types of stretchings, clearly distinguished. Passing to the GO5 spectrum is accompanied by the expected loss of band III, which is associated with carbonyl necklace on the domain edge atoms, while the first two types of modes, especially those underlying band I, remain practically unchanged. The next transition to GO6 concerns the interrelation between *ν sp*^3^C-C and *ν sp*^3^C-C(O) modes, leading to a considerable smoothing of the latter, due to which a highly selective set of them providing a distinguished band I in GO4 and GO5 is transformed in a structureless background in the GO6 spectrum. The situation is similar to that previously discussed for the GO1–GO3 spectra, which allows suggesting that band I should be attributed to the *ν sp*^3^C-C(O) stretchings and its appearance is tightly connected with the epoxy coverage of the graphene domain basal plane, while a total hydroxylation of the plane does not suit the band origin.

## 6. Monitoring the *sp^2^*-to-*sp^3^* Transformation of the Covalent C-C Bonds for DTs of the Third Approach

We consider the formation of GO as a gradual transformation of the *sp*^2^ benzenoid carbon structure of a bare graphene domain into the *sp*^3^ cyclohexanoid configuration of graphene oxide in the presence of particular oxygen reagents. The ACS spin-density algorithmic approach allows monitoring the oxidation gradually. The action can be clearly presented with a set of data that include the relevant DT equilibrium structures, the distribution of their C-C covalent bonds over lengths, and their virtual Raman spectra related to each oxidation step. Skipping the first stage of the oxidation, which concerns the edge atoms of the basic domain (5,5) NGr and taking thus formed necklaced graphene molecule C_66_O_22_ (GO4_00 below) the reference point of the basal-plane-carbon oxidation, Figure 7 exhibits results concerning the first four steps of the chemical action occurred on the domain basal plane. The choice of each subsequent-step target bond is governed by the ACS spin-density algorithm in the format of the *up* and *down* basal plane access.

As seen in the figure, the location of the first epoxy group on the basal plane (GO4_01) results in a significant reconstruction of at least 20 bonds of the reference *sp*^2^C-C bonds pool, which is caused by the expected elongation of bonds 41, 43, 44, and 48. It should be noted that the bond number is retained and fixed throughout all the subsequent steps. Confirming that Raman scattering produces a spectral signature of the C-C bond configuration [4], in comparison with the reference GO4_00 one, the spectrum reveals a sharp response to the change in the bond structure. This feature accompanies the addition of each next epoxy group. Thus, the second epoxy group addition reveals the bond transformation of GO4_02 more markedly and this effect increases in the subsequent steps. Since the observed *sp*^2^-to-*sp*^3^ transformation of the C-C covalent bond structure is accompanied by unavoidable mechanical stress, the compensation of the latter causes not only the expected elongation of the reference *sp*^2^C-C bonds due to their *sp*^3^ transformation, but also a considerable shortening of some of the remaining ones to protect the whole carbon body from destruction.

Figure 8 presents the completion of the oxidation monitoring. The last steps are followed with a severe decrease of each ACS value, which is zeroing at the 17th step. Simultaneously, the bond length dispersion $\Delta l_{C-C}$ drastically decreases, which causes a remarkable structure ordering of the bonds pool of GO4_17 with respect to preceding ones. The bond distribution reveals two types of bonds, first of which concerns the main massive of single *sp*^3^C-C ones of 1.50 ± 0.01 Å in length and the second is related to five highly shortened still remaining untouched *sp*^2^C-C bonds of 1.35 ± 0.001 Å. The shortening of any *sp*^2^C-C bond length below 1.395 Å leads to a complete zeroing of its atoms chemical activity expressed with $N_{DA}$ [42], due to which the current reaction is terminated. Such a quasi-threshold ordering of the C-C covalent bonds structure, which is accompanied by $N_{DA}$ zeroing and, as a consequence, with the termination of further derivatization, is typical for *sp*^2^ nanocarbons. This was observed in all cases of virtual syntheses of hydrides and fluorides of C_60_ fullerene (see the relevant references in [42]) as well as of hydrides and oxides of graphene domains [3].

As seen in Figure 8c, a dramatic change in the Raman spectra occurs at the 17th step of the addition of the epoxy group to the basal plane of GO4 simultaneously with its structural transition described above. The spectrum becomes much more structured and the manifestation of bands II and I is clearly evidenced. Further oxidation can be performed manually only by randomly placing epoxy groups over five remaining *sp*^2^C-C bonds. As seen from the figure, the distribution of bonds over lengths of the last-step GO4 (GO4_22), involving a full set of attached epoxy groups, practically does not change in this case. However, the former double *sp*^2^C-C bonds become single *sp*^3^ ones and band II disappears from the final Raman spectrum. Thus, simultaneous monitoring of the per-step synthesis of the oxide by the C–C bonds pool and Raman spectra allows us to conclude that the I-II doublet in the oxide spectrum is due to the establishment of a balanced structure of highly ordered pools of *sp*^3^ and *sp*^2^ C-C bonds, which keeps the carbon skeleton the most stable and undestroyed. Meeting both requirements is accompanied by an obvious structural ordering of the carbon skeleton, which manifests itself in a small and almost vanishing dispersion of *sp*^3^ and *sp*^2^ C-C bonds by length, respectively.

Confirmation of what has been said above follows from Figure 9, which shows the vibration spectra of a set of DTs discussed in this paper. The spectrum in panel (a) belongs to the bare graphene domain (5,5) NGr and represents the full spectrum of vibrations provided with a total pool of *sp*^2^C-C bonds only. As seen the spectrum is clearly divided into two parts related to torsions, bendings, breathings, etc (0–1000 cm^−1^) and *ν sp*^2^**C**-C stretchings (1000–1800 cm^−1^). The spectrum in panel (b) belongs to the same domain in the necklace of 22 oxygen atoms, which corresponds to one of the configurations typical to rGO [4]. As can be seen in the figure, the spectrum preserves a two-part view, but two peculiarities appear in the previous stretching part: a group of *ν sp*^2^**C=O** stretchings at 2100 cm^−1^ and a particularly high frequency *ν sp*^2^C-C(O) ones at ~1750 cm^−1^. As discussed earlier, the appearance of this group of *ν sp*^2^C-C stretchings should be attributed to the presence of oxygen atoms in the C-C bonds surrounding.

The spectrum in panel (c) corresponds to the case when the oxidation of the domain (5,5) NGr is terminated because of the ACS zeroing. The spectrum is fully reconstructed, particularly in the stretching part over 1000 cm^−1^. It is natural to connect the change with a largely expanded stretching pool, which now involves *ν sp*^3^C-OH, *ν sp*^3^C-O-C, *ν sp*^3^C-C, *ν sp*^3^C-C(O), *ν sp*^2^C-C, and *ν sp*^3^C=O modes. In accordance with Table 1 we believe that the first two modes form the band in the 1000–1200 cm^−1^, the pool of *ν sp*^3^C-C ones is responsible for the (1200–1600 cm^−1^) band, band I at 1750 cm^−1^ is attributed to the *ν sp*^3^C-C(O) modes, band II—to *ν sp*^2^C-C ones, and band III—to *ν sp*^3^C=O.

The spectrum in panel (d) is related to (5,5) graphane [58] virtually designed on the basis of graphene domain (5,5) NGr. Its stretching part above 1050 cm^−1^ involves the only band related to *ν sp*^3^**C-C** modes and is accompanied with a well-distinguished band at 950 cm^−1^ that corresponds to ***δ***
*sp*^3^**C-H** bendings. This spectrum structure is typical for a large class of hydrocarbons, graphane crystal [59], and diamonds [62]. Comparing spectra in panels (c) and (d) makes it possible to clearly reveal the difference in the heteroatoms influence on the vibrational modes of the covalent C-C bond dynamics of the carbon carcass of graphene domain polyderivatives.

To check how common the proposed assignment of the Raman spectrum of GO4_17 presented in Figure 9c is, let us come back to the spectra of DT’s GO1 and GO3 presented in Figure 4b. According to the assignment, the presence of bands I and II in these DTs’ spectra should be related to particular features of the C-C covalent bond pools of the relevant carbon carcasses. Figure 10 shows the C-C bonds’ length distributions related to these DTs. Both DTs correspond to incomplete oxidation, which is terminated on the 17 and 38 steps of the reagent placed on the basal plane of the pristine domain, respectively, because of ACS zeroing [43]. As seen in the figure, in both cases this reaction step is accompanied by the formation of a few shortened *sp*^2^C–C bonds of 1.350 ± 0.003 Å in length. It is these low dispersed bonds, which cause the appearance of band II in the virtual Raman spectra of GO1 and GO3, once band G in experimental GO spectra. As for the arrays of *sp*^3^ C-C bonds, they are naturally well structured in both cases and constitute 1.505 ± 0.032 Å and 1.548 ± 0.043 Å, respectively. However, the band I appearance evidently depends on how well *ν sp*^3^C-C(O) modes are differentiated from the total massive of *ν sp*^3^C-C ones. As seen in Figure 4b and Figure 6b, in the case of GO1, GO2, GO4, and GO5 these modes are well separated and form rather narrow-band frequency massifs, which provides well-distinguished bands I in the DTs’ Raman spectra. In contrast, in the case of GO3 and GO6, ***ν***
*sp*^3^**C**-C(O) modes form a broad-band frequency region which overlaps with a wide spectrum of the main array of ***ν***
*sp*^3^**C**-C modes, thus being lost in a structureless background of these modes. Therefore, as was shown earlier and confirmed in this work, the nature of the chemical agent responsible for oxidation is the main factor influencing the shape of the Raman spectrum of GO in the region of band I. The band I presence, the D band in empirical spectra, indicates that atomic oxygen is the main chemical agent that provides a coating of the basal plane of the oxide graphene domain with epoxy groups. The presence of hydroxyl groups in the basal plane is not completely ruled out in this case, but their number is small, since, otherwise, band I would be absent.

## 7. Discussion on the Digit Twins Concept and Conclusive Remarks

This paper presents the first results of considering a unique spectral phenomenon—the identity of Raman spectra of parental and reduced graphene oxides, an explanation for which has not been found to date. As time has shown, the answers cannot be obtained empirically, in contrast to which a wide chemical and structural modification, possible within the DTs concept, hopes to achieve the goal. According to Scheme (1), the concept application succeeds when certain requirements are met. Those are the following: (1) DTs form a large set of self-constitent independent assets, which allows a wide comparative study of the results obtained; (2) virtual device properly reflects and reconstitutes the characteristic features of the objects under consideration; (3) the obtained IT product represents a set of data that best fit the empirical ones under discussion. The current study was performed to meet these demands.

Once independent, DTs are not arbitrarily designed but are subbordinated to the absorption of all the knowledge gained on real objects. From this viewpoint, GO and rGO form an inseparable DTs’ chain connecting three products. The Scheme (2) is as follow:(2)Bare graphene domain ↔ Graphene oxide ↔Reduced graphene oxide

Here the sign ↔ means that the chemical modification, which interconnects each of the corresponding pairs, is reversible. All the chain members are highly complicated chemical objects, starting from polytarget radical bare graphene domains and continued and finished with these domain polyderivatives. The latter are of two types characterized by either total involvement (GO) or total ignoring (rGO) of the domain basal plane atoms in the oxidation additionally to edge domain atoms fully involved in both cases. Getting answers pointed out the above results in a quite large number of DTs to be considered that achieves in the current case a few hundreds.

The virtual device is faced with the following tasks: (1) provision of a transparent design of the required DTs, (2) performance of the appropriate computational synthesis of the DTs, and (3) calculation of IR absorption and Raman scattering spectra of the relevant DTs. The solution to the first problem is facilitated by the radical nature of all products of the chain in Scheme (2). This made it possible to lay the spin molecular theory of *sp*^2^ nanocarbons [42] into the foundation of the device and use the ACS spin-density algorithm for the design of their polyderivatives. The corresponding virtual synthesis-design of stable polyderivatives of graphene domains was carried out using the CLUSTER-Z1 software [6], which implements the semi-empirical restricted and/or unrestricted Hartree-Fock approximations. The high efficiency of the techniques as well as the excellent set of parameters, worked out over decades of successful use, made it possible to obtain reliable DTs, each consisting of one to several hundred atoms.

To cope with the third task, a virtual vibrational spectrometer HF Spectrodyn was used [48], based on the same semi-empirical Hartree-Fock approximation using either restricted or unrestricted versions when solving the vibrational problem [7] in the absence or presence of radical properties in the DTs under consideration, respectively. The obtained IT products present one-phonon harmonic spectra of IR absorption and Raman scattering. Its deviation from the empirical analogues concerns the lack of anharmonicity and a considerable blue shift. The former is highly important empirically, leading to a remarkable change in the optical spectra with respect to harmonic ones [63]. Nevertheless, the main features of both IR and Raman empirical spectra of the molecular *sp*^2^ nanocarbons are of harmonic origin. Accordingly, harmonic IT product reproduces the latter quite properly. As for the unavoidable blue shift of virtual harmonic frequencies [55], it is quite considerable and constitutes ~200–300 cm^−1^ in the 1000–3500 cm^−1^ region. However, it is the same for all the studied DTs and can be ignored when comparing virtual data of the species but should be taken into account at the final stage of comparing virtual and experimental data.

GO is the most complex species in Scheme (2) and its consideration requires a thorough analysis of two other chain members. Accordingly, the first part of the research on the chain program was performed for graphene domains, while the second was particularly devoted to rGO. It was found that IR spectra of bare domains are extremely weak while intense Raman spectra are a bright signature of the *sp*^2^C-C covalent bonds and are provided with **ν**
*sp*^2^**C-C** stretchings [64]. These spectra shape highly depends on the linear size of the domain, replacing a broad multiband structure at the domain size below 10–15 nm with a strongly narrowed single G band at a bigger size. The transition is connected with changing the molecular character of the domain vibrations into phonons of graphene crystal when the domain size is approaching or exceeds the free path of optical graphene phonons $L_{ph}$~15 nm [31,65]. Important to note that none of the individual domains is characterized by a remarkable D-G doublet structure of Raman spectra. The latter originates when the domains are packed in layers. Unfortunately, this cannot be checked experimentally since bare graphene domains do not exist.

The transition from bare graphene domains to necklaced ones of various rGOs still leaves us in the field of *sp*^2^C-C covalent bonding [4], but the covalent bonds pool changes significantly. An additional set of *sp*^2^C-A bonds, formed upon the termination of highly active edge atoms of the domain with heteroatoms A, is added to the set of *sp*^2^C-C bonds inherent in the bare domain. The two groups of bonds are differently active in optical vibrational spectra. Thus, *sp*^2^C-A bonds are mainly responsible for IR absorption, providing strong variability of IR spectra with respect to the rGO chemical origin. In contrast, the domain *sp*^2^C-C bonds determine the main pattern of the rGO Raman spectra which are dependent on the rGO origin only slightly. It was found that, as in the case of bare domains, individual rGOs are not characterized with standard D-G doublet Raman spectra. As previously, the latter is a signature of the domains multilayer stacking. The intensity of D band increases when the number of layers grows up to 4–5 and then markedly slows down when the stack thickness exeeds ~15 nm [66]. Particular doublet patterm of the spectra strongly depends on the linear size of the rGO DT and is transformed from a broad-band one to a narrow-band with a drastic domination of G band when the size of the graphene domain exceeds the free path of graphene optical phonons $L_{ph}$~15 nm. Therefore, the characteristic D-G doublet structure of rGOs’ Raman spectra is of structural origin, evidences the stacked nature of the corresponding solid structure and manifests the formation of special dynamic single *sp*^3^C-C bonds between atoms located in adjacent layers. Thus, the author’s analysis of the structural and chemical-content data, particularly Raman spectra, of the cosmogenic carbon, brought to the Earth by the Chelyabinsk meteorite [67], leads to a confident conclusion about a graphite-like stacked structure of this carbon with a stack thickness from ~10–15 nm to more.

In contrast to rGO, whose heteroatom chemical modification is concentrated only in the domain circumference, the GO formation applies to the entire domain, including the carbon atoms of the basal plane as well. Thus, if in the case of rGO *sp*^2^ hybridization of the electronic structure of carbon atoms is retained not only in the basal plane, but sometimes takes the place of the valence-saturated edge atoms as well, the hybridization of the GO carbon atoms becomes *sp*^3^ one throughout the domain structure. This leads to the replacement of benzenoid planar structures of rGO domains by cyclohexanoid puckered structures of GO ones, which increases the thickness of the domain from 0.335 nm to 0.479 nm and this alone leads to an increase in the interlayer distance in solid GO, in contrast to rGO. Thus, complete oxidation of the domain is accompanied by an *sp*^2^-to-*sp*^3^ transformation of the carbon carcass of the pristine graphene domain.

The formation of almost planar rGO domains does not raise any special questions, since the diversity of their chemical compositions, affecting only a few atomic percent of its structure, is quite reliably predictable and explained by the difference in chemical reactions that provide the formation of heteroatom necklaces in each specific case (see [4] and references therein). In the case of GO, the prediction of its structure is more problematic, since there is always a possibility of participation of different oxygen-containing groups, on the one hand, and the location of the oxidants in different parts and on different sides of the basal plane of the original graphene domain, on the other. Even 10 years ago, considering this problem [43], we abandoned the simple drawing of a possible hypothetical structure and, realizing the inevitable transformation of the initial *sp*^2^ C-C covalent bonds into *sp*^3^ ones at each oxidation act, turned to the virtual synthesis of GO using the spin-density algorithm that tracks the chemical activity of the initial carbon atoms by a quantitative way. The models synthesized at that time were used in the present work as one of the possible DT sets, which gave an unexpected result: the Raman spectra of two of them (GO1 and GO2) have a characteristic structure of a doublet of bands I and II located in the same spectral region as D and G bands in the spectra of rGO. In contrast to rGO, bands I and II are characteristics of individual GO domains and cannot be attributed to the domain stacking.

By changing the initial conditions and slightly correcting the format of the governing algorithm application, we managed to virtually synthesize a large number of GO DTs reflecting various oxidation states, which allowed getting the feature explanation. It was found that, as in the case of rGO, the GO structure retains two parts associated with the difference in the oxidation processes occurring at the edge atoms of the pristine graphene domain and in its basal plane. The *sp*^3^C-A bonds, related to either edge or basal-plane carbons and formed in the course of the paternal graphene domain oxidation, are different because of a particular topochemistry of the domain derivatization [43]. A large variety of GO DTs has allowed for the tracing of the role of these bonds in optical vibrational spectra. Similar to rGOs, *sp*^3^C-A covalent bonds related to edge domain atoms dominate in the GOs’ IR spectra. At the same time, those related to the domain basal-plane atoms bring additional details to these spectra, making it possible to distinguish between oxides with the same necklace of heteroatoms around the edge atoms while differing in the composition of oxidants located in the basal plane of the domain. A particular list of general frequency kits was suggested to provide a reliable chemical-content interpretation of the GOs’ real IR spectra.

Concerning the band-doublet structure of the GO Raman spectra, it was found that it is characteristic of a predominant covering of the graphene domain basal plane with epoxy groups. A similar covering with hydroxyls is accompanied by the loss of band I. This reaction on the substitution of atomic oxygen oxidants with hydroxylic ones seems to be unexpectedly sharp since the band is provided with excitation of *sp*^3^C-C stretchings that evidently form the main pool of vibrational modes in both cases. However, a large variety of the DTs considered allowed establishing that not all the *sp*^3^C-C covalent bonds are involved in the Raman scattering, but only *sp*^3^C-C(O) ones that are in touch with epoxy units. Therefore, the presence of band I in virtual as well as of band D in empirical Raman spectra of GOs evidences that basal planes of graphene domains of the bodies’ basic structural units are covered with epoxy groups predominantly.

Conditions, providing the observation of G band in the GO’s empirical Raman spectra and band II in virtual ones, occurred to be more rigid. A set of particular DTs, supplying monitoring of successive oxidation in basal planes of paternal domains, allowed fixing the band II appearance at the moment of the spin-induced termination of oxidation. The achieved oxidation extent of 80–90% is determined by zeroing the atomic chemical susceptibility of *sp*^2^carbon atoms due to the shortening of *sp*^2^C-C covalent bonds below their critical length [42]. The shortening is stimulated to compensate for the mechanical stress which arises in the course of the *sp*^2^-to-*sp*^3^ transformation of the C-C covalent bonds of the GO domain because of the difference in the length of *sp*^2^- and *sp*^3^ C-C bonds as well as the relevant C-C-C valence angles. These remaining *sp*^2^C-C covalent bonds generate the relevant *sp*^2^C-C stretchings, which provide the appearance of the band II in virtual and band G in empirical Raman spectra of graphene oxides. Therefore, clearly distinguished D-G doublet structure of GOs’ Raman spectra is of chemical origin and manifests a spin prohibition of 100% oxidation of the parent graphene domain.

Thus, the simultaneous participation of *sp*^2^*-* and *sp*^3^C-C covalent bonds in the scattering of light in both cases has been the reason for the identity of Raman spectra of rGOs and GOs. These bonds are characterized by sets of stretchings with a wide frequency range. This circumstance undoubtedly favors the fact that the vibrational structures of Raman spectra of the two substances can significantly overlap. However, in the virtual and empirical spectra, an almost complete coincidence of the positions of the bands’ doublet is observed. As can be seen from the conducted virtual experiment, in both cases, high-frequency modes of both types are involved in the Raman scattering in both cases. Under the conditions of the revealed difference in the origin of the relevant pairs of covalent bonds, the question of the particular role of the high-frequency fragments of stretching modes in the formation of the Raman spectrum deserves further study.

The proposed analysis of the vibrational spectra of rGO and GO is based on the important concept of the radical nature of bare graphene domains and rGO. Many spears have been broken in discussions that cast doubt on the radical nature of graphene materials. The author would like to hope that the described study will allow for the rejection of the last remaining doubts and will force researchers to work with materials familiar to them in a new way.

## Figures and Tables

**Figure 1 nanomaterials-12-04209-f001:**
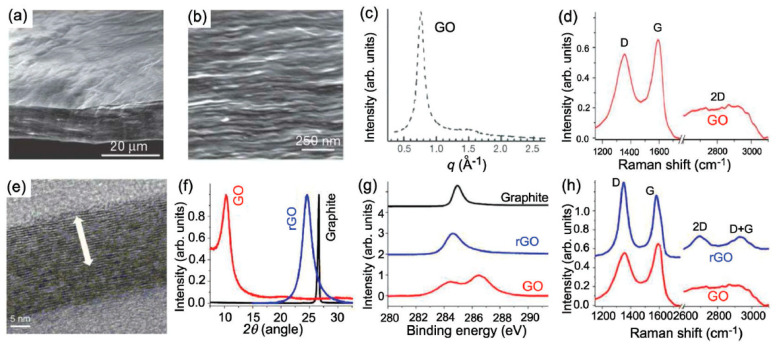
(**a**) Low- and (**b**) high-resolution SEM side-view images of a 10 mm-thick GO paper. (**c**) X-ray diffraction pattern of the GO paper sample. (**d**) Raman spectrum of a typical GO paper. (**e**) Cross-section TEM images of a stack of rGO platelets. (**f**) Powder XRD patterns of graphite, GO, and rGO. (**g**) XPS characterization of rGO platelets. (**h**) Raman spectra of rGO (blue) and the GO reference sample (red). Reproduced from [13].

**Figure 2 nanomaterials-12-04209-f002:**
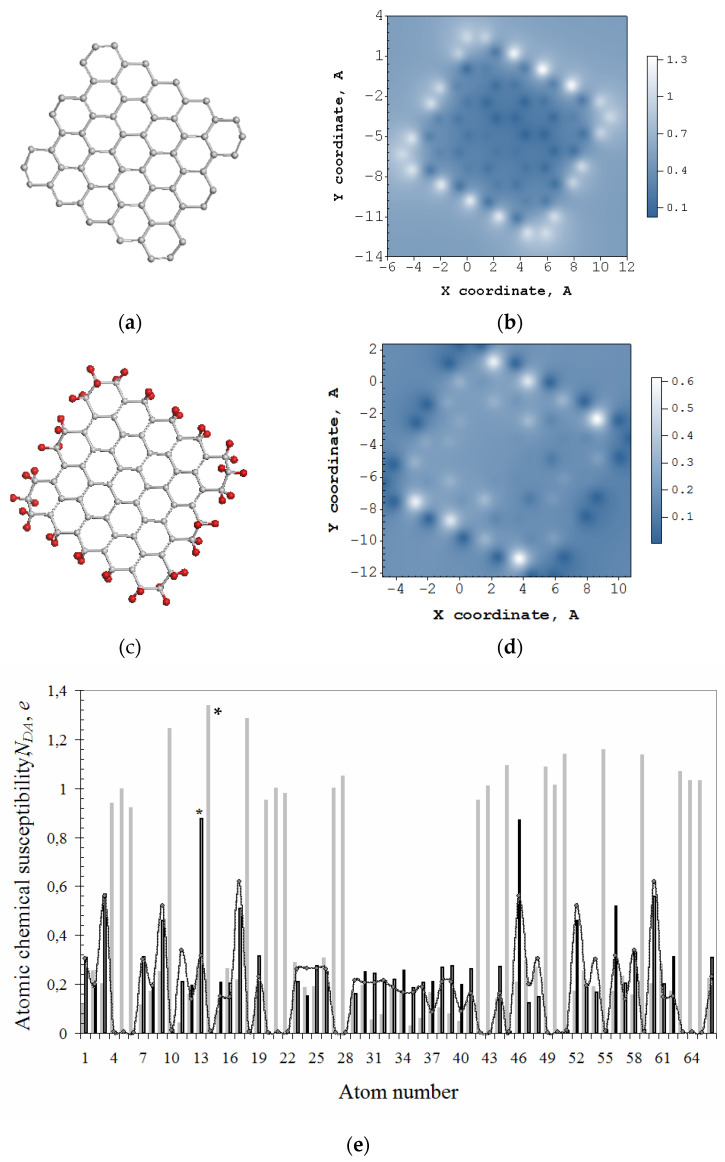
Equilibrium structures of bare graphene domain (5,5) NGr (C_66_) (**a**) and free-standing that one, but terminated by two hydrogen atoms per each edge one (C_66_H_44_) (**c**). ACS $N_{DA}$ image maps over atoms of these molecules in real space (**b**,**d**) as well as the above ACS distribution according to the atom number in the output fil€ (**e**). Light gray histogram plots ACS data for C_66_, while the black one and curve with dots are related to C_66_H_44_. The curve maps free valence distribution over atoms. Scale bars match $N_{DA}$ values. UHF AM1 calculations.

**Figure 3 nanomaterials-12-04209-f003:**
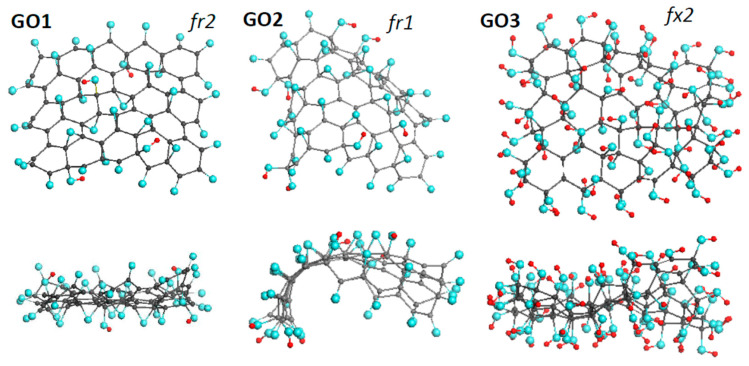
Face and side view of equilibrium structures of virtually synthesized graphene oxides GO1 and GO2, both of C_66_O_40_H_4_ content, and GO3 of C_66_(OH)_74_. Gray, blue, and red balls mark carbon, oxygen, and hydrogen atoms, respectively. UHF AM1 calculations.

**Figure 4 nanomaterials-12-04209-f004:**
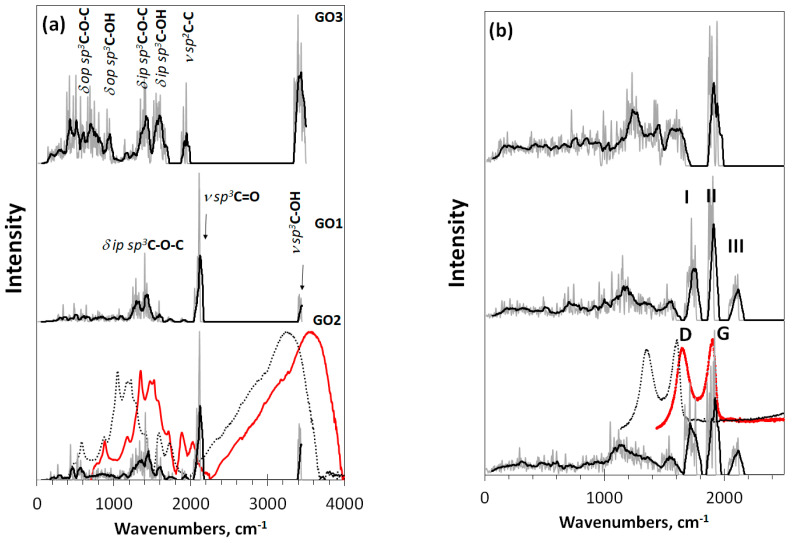
Virtual one-phonon spectra of (**a**) IR absorption and (**b**) Raman scattering of digital twins GO1, GO2, and GO3. Spectra plottings are accompanied by trend lines, corresponding to 50-point linear filtration. RHF AM1 calculations. Dotted and red plottings present original and blue shifted on 300 cm^−1^ experimental spectra of GO manufactured by AkKo Lab [54], respectively.

**Figure 5 nanomaterials-12-04209-f005:**
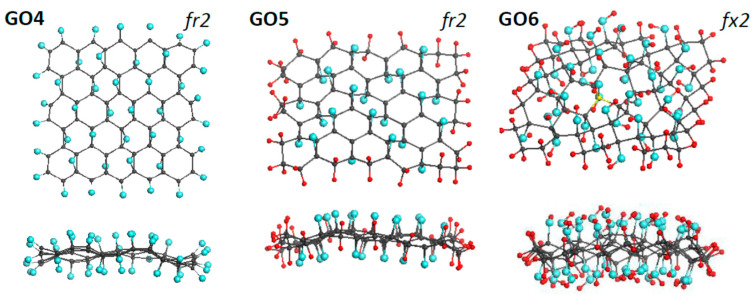
Face and side view of equilibrium structures of virtually synthesized monochrome graphene oxides GO4 (C_66_O_44_), GO5 (C_66_H_44_O_22_), and GO6 (C_66_O_44_H_88_). Gray, blue, and red balls mark carbon, oxygen, and hydrogen atoms, respectively. UHF AM1 calculations.

**Figure 6 nanomaterials-12-04209-f006:**
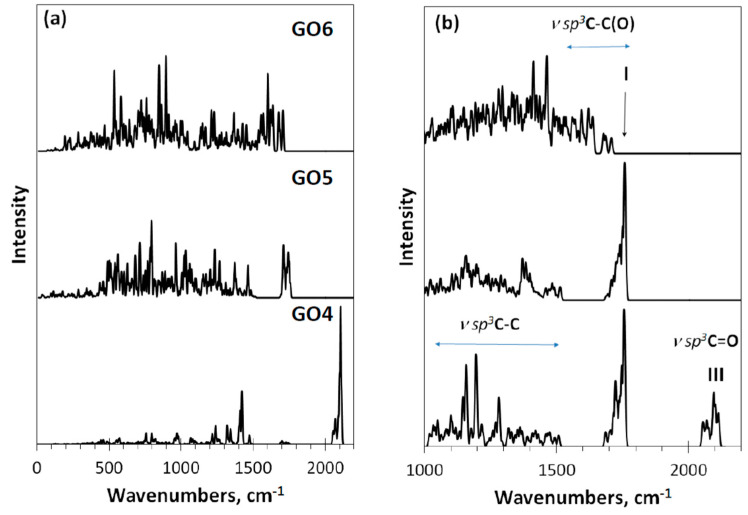
Virtual one-phonon IR absorption (**a**) and Raman scattering (**b**) spectra of digital twins GO4, GO5, and GO6. RHF AM1 calculations.

**Figure 7 nanomaterials-12-04209-f007:**
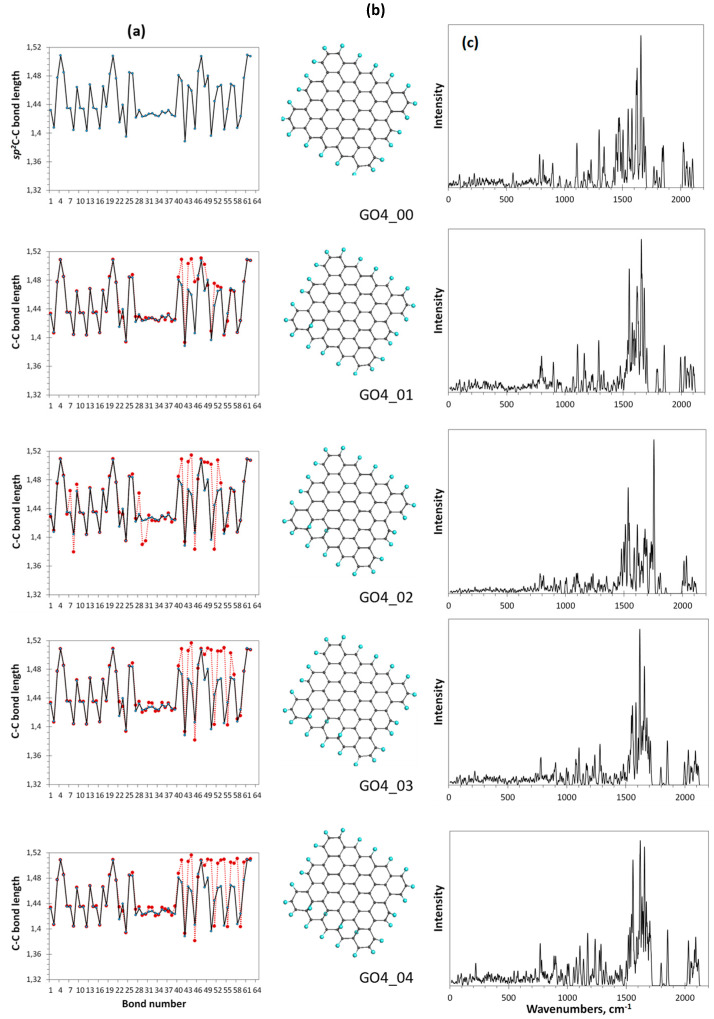
Monitoring of the *sp*^2^*-*to*-sp*^3^ C-C covalent bond transformation at first steps of the oxidation of basal-plane carbon atoms of the reference GO4_00 TD. (**a**). TD’s C-C bond length distributions (reference and step current presented with solid black and dotted red plottings, respectively). (**b**). TD’s equilibrium structures. (**c**). TD’s Raman spectra. UHF AM1 calculations.

**Figure 8 nanomaterials-12-04209-f008:**
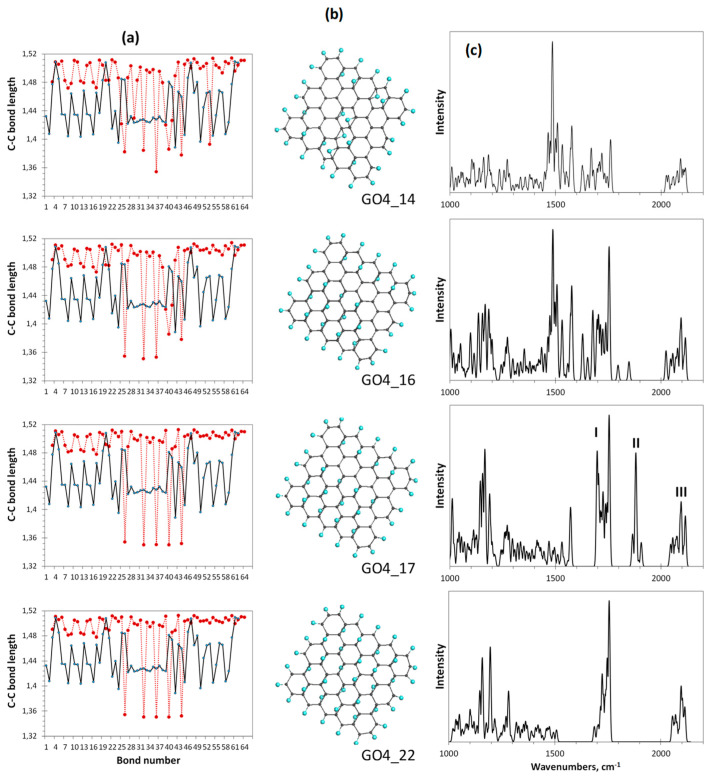
Monitoring of the *sp*^2^*-*to*-sp*^3^ C-C bond structure transformation the at final steps of the oxidation of basal-plane carbon atoms of GO4. (**a**). TD’s C-C bond length distributions (reference and step current presented with solid black and dotted red plottings, respectively). (**b**). TD’s equilibrium structures. (**c**). TD’s Raman spectra. UHF AM1 calculations.

**Figure 9 nanomaterials-12-04209-f009:**
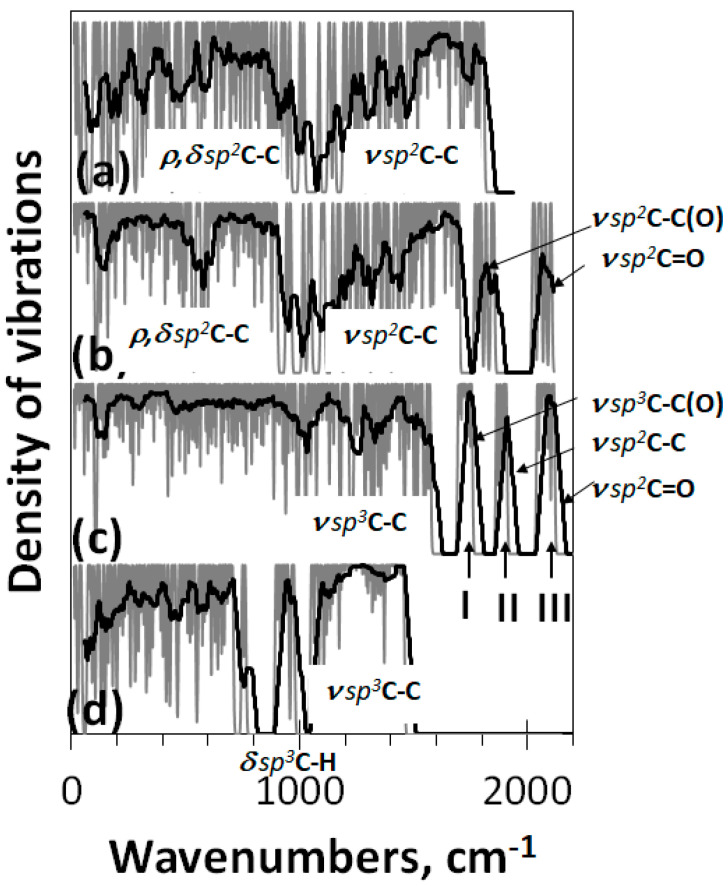
Virtual one-phonon vibration spectra of bare (5,5)NGr domain C_66_ (**a**) and its polyderivatives: reduced graphene oxide C_66_O_22_ (**b**), graphene oxide GO4_17 C_66_O_39_ (**c**), and graphene hydride (nanographane [60]) C_66_H_88_ (**d**). UHF (**a**,**b**) and RHF (**c**,**d**) AM1 UHF calculations.

**Figure 10 nanomaterials-12-04209-f010:**
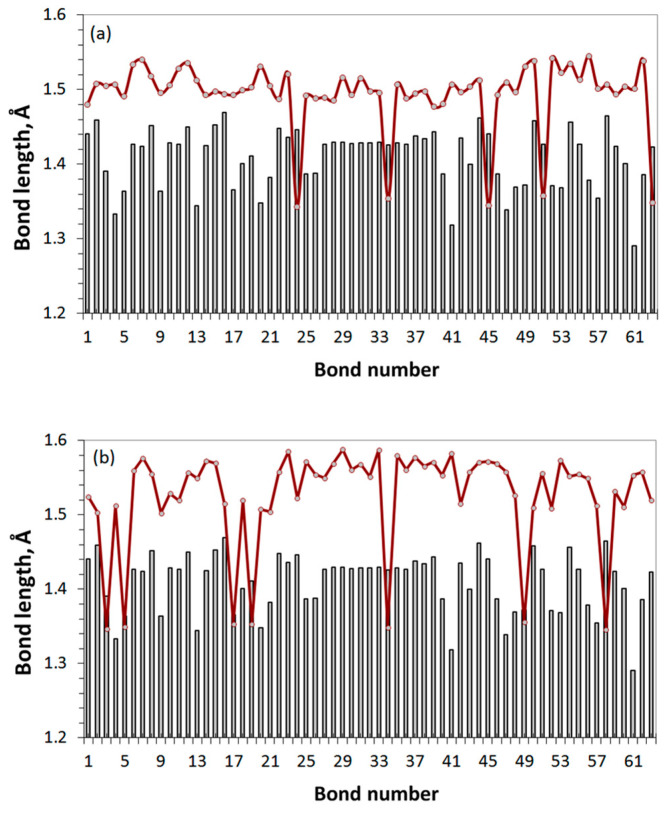
The *sp*^2^*-*to*-sp*^3^ transformation of the C-C covalent bonds of the (5,5) NGr domain in consequence of the successive hetero-oxidant action. The final-stage oxidation of (**a**) GO1 and (**b**) GO3 (see text for details). Gray histogram is related to the pristine domain.

**Table 1 nanomaterials-12-04209-t001:** General frequencies kits to suitably interpret vibrational spectra of rGOs and GO ^1^, cm^−1.^

Spectral Areas									
Experiment ^2^	300–1000	1000–1200	1200–1300	1300–1500	1500–1600	1600–1700	1800–1900	2600–3000	3000–3600
rGO	*δ op*^4^, *δ ip*^5^, ρ torsions *sp*^2^**C-O-C** and *sp*^2^**C-O**H *δ op sp*^2^**C-C-C** ^6^ *δ ip*, puckering, ring breathing, *δ* trigonal *sp*^2^**C-C-C** ^6^ , collective vibrations of domain atoms ^7^	*ν sp*^2^**C-O-C** in cyclic ether, aggregated cyclic ether and acid anhydride,	*ν sp*^2^**C-O**H, in lactone, hydroxyl pyran and acid anhydride	*δ ip* *sp*^2^**C-O**H, *ν sp*^2^**C-O-C** in cyclic ether and acid anhydride *δ ip* **O-C=**O in acid anhydride	*δ ip* *sp*^2^**C-O**H, *ν sp*^2^**C-C**	*ν sp*^2^**C=O** in acid anhydride and lactone, aggregated cyclic ether with lactone pair, pairs of lactones	*ν sp*^2^**C=O** in *o*-quinone, COOH	*ν sp*^3^C-**O-H** in COOH *ν sp*^3^**C-H**	*ν sp*^3^C-**O-H**
GO	*δ op*^4^, ρ torsions of *sp*^3^**C-O-C** and *sp*^3^**C-O**H, *δ op sp*^3^**C-C-C**^8^ *δ ip*, puckering, cyclohexane ring breathing, *sp*^3^**C-C-C** ^8^	*δ op* *sp*^3^**C-O**H, *ν sp*^3^**C-OH**, *ν sp*^3^**C-O-C**, *ν sp*^3^**C-C**^9^	*δ ip* *sp*^3^**C-O-C**, *ν sp*^3^**C-O-C** In pairs of cyclic ether and lactones. *ν sp*^3^**C-C**^9^	*δ ip* *sp*^3^**C-O**H, *ν sp*^3^**C-OH**, *ν sp*^3^**C-O-C**, *ν sp*^3^**C-C**(O) ^10^	*ν sp* ^2^ **C-C**	*-*	*ν sp* ^3^ **C=O**	*-*	*ν sp*^3^C-**O-H**
Virtual data ^3^	300–990	1280	1410–1550	1560–1660	1800–1900	-	1950–2100	-	3420

^1^ Greek symbols *δ* and *ν* mark the molecule bendings and stretchings, respectively. ^2^ The assignment of frequencies in the experimental spectra is based on the papers [49,50,51,52,53]. ^3^ Obtained in the current study. ^4^ Out-of-plane bendings. ^5^ In plane bendings. ^6^ Benzene molecule data [56]. ^7^ Virtual data for nanographene [57]. ^8^ The author approximated suggestions for GO. ^9^ Virtual *sp*^3^**C-C** stretchings of graphane [58]. ^10^ Virtual *sp*^3^**C-C(O)** stretchings affected with the oxygen presence nearby (current paper).

## Data Availability

Any data or material that support the findings of this study can be made available by the corresponding author upon request.

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
