# Peer review of "Digital Twins Solve the Mystery of Raman Spectra of Parental and Reduced Graphene Oxides"

_nanomaterials, 2022, doi:10.3390/nano12234209_

Round 1

Reviewer 1 Report

This paper is very disappointing, is over 29 pages long, and with unproved claims and forgiving major issues. Where are the resonance effects in the claim (line 19) "It was established that in both cases the D-G doublets owe their origin to the sp 3 -sp 2 C-C stretchings, respectively." I expect to have the absorption of each phase at the considered energy and I was unable to find it. Raman spectroscopy of sp2/sp3 requires to be properly treated. Wavelength dependence has been tracked for a long time: see Profeta, M., & Mauri, F. (2001). Theory of resonant Raman scattering of tetrahedral amorphous carbon. Physical Review B, 63(24), 245415. Double resonance with 2 approaches is nowadays accessible in so many cases: Torche, A., Mauri, F., Charlier, J. C., & Calandra, M. (2017). First-principles determination of the Raman fingerprint of rhombohedral graphite. Physical Review Materials, 1(4), 041001. Here, this crucial point is not discussed while experimentally, it's a key parameter: see for example Sharma, N., Sharma, V., Jain, Y., Kumari, M., Gupta, R., Sharma, S. K., & Sachdev, K. (2017, December). Synthesis and characterization of graphene oxide (GO) and reduced graphene oxide (rGO) for gas sensing application. In Macromolecular Symposia (Vol. 376, No. 1, p. 1700006). Here, we probably switch between two regimes but it should be calculated. I find 20 citations of the author while many references are missing. Please correct that. The choice of the figure in the literature (fig1) is not sufficient. For example, FTIR (see Sharma et al) is very different while only Raman data from other authors are reported here (where I was unable to find the discussion on the relative intensity variation with the excitation energy). The author should focus on what is new, clearly explain the limitations of their model, and add energy dependence. In the present form, I cannot recommend its publication.

Reviewer 2 Report

This manuscript describes modelling of graphene and reduced graphene oxide to try to tease apart the particular structural elements responsible for particular features in the IR and Raman spectra of graphene oxides.  The manuscript describes the use of a set of selectively oxidized graphene models to investigate which spectral features appear under what conditions and correspond to what structural elements.  This is a nice example of how something that is difficult to study experimentally can be modelled in ways that allow the behavior of the complex material to be more readily analyzed.  In this case, the selective oxidation to form different digital twins and comparison of their IR and Raman spectra allow identification of the the D and G bands in the Raman spectra as being related to the sp2 and sp3 hybrid carbons in the different graphene structures. 

The differences in the spectra are clear in some cases, but less obvious in others.  There are a couple of places where it is not obvious what changes the author is refering to when they describe the appearance of the spectra.  Better labeling of the figures, with additional markings to highlight changes or important features might help with this.

Table 1 is mentioned multiple times, but does not appear to be present in this version of the manuscript.  Since this is to contain the assignments of the peaks, it is critical that this table be easily referenced.

The discussion regarding what is responsible for the D and G bands in the rGO (dynamic sp3 C-C bonds, on page 22) is not clear.  How is this determined from the observation of these structures in the single layer calculations carried out here?

While there do seem to be compelling results and conclusions which can be reached using this novel set of model structures, the many significant typos, incorrect words, and poor phrasing make it very difficult to understand the main points that are trying to be made.  The manuscript would benefit from significant trimming and editing to tighten up the writing and focus on the main points.  

Reviewer 3 Report

The manuscript entitled Digital Twins solve the mystery of Raman spectra of 2 parental and reduced graphene oxides reports on the very concept of digital twins (DT) that may solve the mystery of Raman spectra of two parental and reduced graphene oxides. DT presents a new trend in virtual material science,  common to computational approaches. Digital twins, virtual device and intellectual product present main constituents of the DT concept, which are considered for complex graphene oxide (GO) and reduced graphene oxide (rGO) materials Raman spectra as a case study. The authors discussed the D-G-doublet Raman spectra of GO and rGOs. Through DTs, different aspects of the GO and rGO structure and properties are virtually synthesized using spin-density algorithm emerging from the Hartree-Fock (HF) approximation. It was established that in both cases the D-G doublets owe their origin to the sp3 -sp2 C-C stretchings, respectively. Thus, multilayer packing of individual rGO molecules in stacks, and spin-influenced prohibition of the 100% oxidative reaction, in the second, provided an exclusive identity of species in Raman spectra. While this study was an interesting and non-traditional way of thinking about the problem of the structure of GO and rGO via identifying the signatures in Raman spectra, there are numerous drawbacks to the study.

1. The presentation of the manuscript appears to be a chapter, not a research paper. It requires re-organizing the whole manuscript to be presentable relevant to the scientific community.

2. It requires heavy editing throughout the manuscript, too many to enlist them here. 

3. The Schemes need to be expanded or re-organized as they are not apparent from a scientific point of view.

4. The authors should consider the possible structures of GO and rGO from the existing literature and highlight the novelty in their study.

5. The paper can be reconsidered only if these major revisions are taken into account.

Round 2

Reviewer 1 Report

In my previous review, I explained my point of view.
Unfortunately, the author declined to answer any of my questions, my problem is not just details, it corresponds to the whole philosophy of the article. I gave examples to explain to the author what is feasible and they were just rejected without any consideration.
I have explained double resonance is now well-admitted in the community of carbon, and why the present approach is more suitable for FTIR than for Raman.
Ref 13 used for fig 1 explains that in detail.
A number of self-citation is maybe required as few people are audacious enough to treat the problem with such a simplification level but this is not satisfactory.
Why not cite Sharma (Sharma et al, Macromolecular Symposia,376, 1, 1700006) to have FTIR? Still a mystery.
How real are Raman spectra obtained? It's not explained if the bond polarizability model is employed or if just DOS is considered. The limit of the model should be clearly detailed. Because it's not a good idea to give a reference (ref 7) where it's not clear from the conclusion if Raman calculation is available.
The title is for me a trickery as I expect really more, not just a simple mechanical simulation.
I have checked and nowhere is written double resonance. This is non-acceptable even if it's not present in the model. It means that the Raman data are not well introduced.
The model is not able to describe peak position (which shifts with exciting wavelength) nor peak intensity (as double resonance is involved in the Raman cross-section)
See for wavelength dependence for example Wroblewska et al, Journal of Physics: Condensed Matter, 29(47), 475201, 2017.
I'm sorry but this article in its present form fails to connect experimental data to simulation.

Author Response

Please 

Reviewer 3 Report

The manuscript reports on solving the mystery of Raman spectra of parental and reduced graphene oxides via digital twins. The digital Twins concept presents a new trend in virtual material science, common to computational techniques. The manuscript concerning the amazing identity of the D-G-doublet Raman spectra of parental graphene oxide (GO) and reduced graphene oxides (rGO) is resolved through digital twins. This is quite an interesting approach to providing the equilibrium structure of the GO and rGO's D-G twin peaks as well as virtual one-phonon IR absorption and Raman scattering spectra. The author elucidated that in both cases the D-G doublets owe their origin to the sp3-sp2 C-C stretchings, respectively. The manuscript is improved to a certain extent after the revision. The author should re-read the paper and revise it accordingly and address mechanical and editorial deficiencies. The paper is acceptable after minor editing for publication. 
